# Antiproliferative Cancer Cell and Fungicidal Effects of Yellow and Red Araçá (*Psidium cattleianum* Sabine) Fruit Extract

**DOI:** 10.3390/foods12234307

**Published:** 2023-11-29

**Authors:** Félix Roman Munieweg, Ana Luisa Reetz Poletto, Jean Ramos Boldori, Cheila Denise Ottonelli Stopiglia, Felipe Barbosa de Carvalho, Sandra Elisa Haas, Nathane Rosa Rodrigues, Márcia Vizzotto, Cristiane Casagrande Denardin

**Affiliations:** 1Campus Uruguaiana, Universidade Federal Do Pampa, BR 472, Km 592, Uruguaiana 97501-970, Brazil; felixmunieweg.aluno@unipampa.edu.br (F.R.M.); anapoletto.aluno@unipampa.edu.br (A.L.R.P.); jeanboldori.aluno@unipampa.edu.br (J.R.B.); cheilastopiglia@unipampa.edu.br (C.D.O.S.); felipecarvalho.aluno@unipampa.edu.br (F.B.d.C.); sandrahaas@unipampa.edu.br (S.E.H.); nathanerodrigues.aluno@unipampa.edu.br (N.R.R.); 2Embrapa Clima Temperado, Pelotas 96010-971, Brazil; marcia.vizzotto@embrapa.br

**Keywords:** Brazilian native fruit, antifungal activity, cancer cell lines, toxicity

## Abstract

Araçá is a native Brazil fruit, and has two morphological types, yellow and red; however, it is still little consumed by the population. Although there are few studies on the araçá fruit, some phytochemical propriety benefits have been described for this plant, such as antioxidant effects. To explore the benefits of araçá fruit, the physicochemical characteristics and in vitro toxicological effects of red and yellow araçá fruit were evaluated. In this work, the toxicity of araçá extracts in NIH/3T3 cell lines, the antiproliferative effects in cancer cell lines (C6, HT-29, and DU149), and the overall antifungal effects were evaluated. The irritant potential of araçá extracts was assessed by the HET-CAM test. The results demonstrated that the fruits are rich in fiber content and showed high phenols content. In addition, the araçá extracts had no present toxicity effects in cell lines; however, the red araçá extracts showed antiproliferative effects in HT-29 cancer cells at 50 mg/mL. The antifungal effects of araçá extract were promising in 23 isolates of *Candida* spp., and both araçá extracts showed no irritant effects. Therefore, this study demonstrated that red and yellow araçá fruit extract has promising biological and pharmacological effects that should be further explored.

## 1. Introduction

The araçá fruit (*Psidium cattleianum* Sabine) is a small berry of the *Myrtaceae* family [1,2] occurring naturally in the area between Bahia and Rio Grande do Sul states (Brazil), easily adapted to a variety of climates [3,4]. Araçá is present in two varieties when ripe, namely yellow and red, which suggests the existence of two morphotypes [5,6]. The yellow and red fruits of “EMBRAPA Clima Temperado” refer to the cultivars “Ya-cy” and “Irapuã”, respectively [7,8]. Fruits can be consumed either in natura or processed as sweets or juices [3,4].

Studies have shown that araçá has a high antioxidants content such as ascorbic acid, phenolic compounds, flavonoids, carotenoids, and terpenoids [1,9,10,11,12,13,14,15]. Some pharmacological activities of araçá leaves extracts have been reported in the literature, such as antimicrobial activity, mainly an antifungal agent [10,13,14,16,17], in addition to the analgesic effect [18], the potential to prevent the tooth enamel demineralization in situ [9], and anti-inflammatory and hepatoprotective effects [19]. Some studies also demonstrate the biological effects of araçá fruit extracts, mainly as antimicrobials in some classes of bacteria and antineoplastic activity in the cell cultures of MCF-7 and Caco-2 [20]. However, it should be considered that most of these studies are concerned with the use of leaves and few studies evaluate the fruits. In addition, there is still limited scientific information available regarding the effects of araçá fruits, in addition to a lack of studies aiming at its use in the form of extract, both for pharmacological and toxicological evaluation.

Based on the previously reported antifungal activities of the araçá leaves, the possible effect of the fruit extract against fungi such as *Candida* spp. is of great pharmacological interest. *Candida albicans* is the main etiologic agent of opportunistic fungal infections [21] and vulvovaginal candidiasis (VVC) is the most common infectious disease of the genital tract, affecting about 75% of women of reproductive age [22,23]. In Brazil, although there are no concrete epidemiological data, the prevalence of VVC is estimated at 18–39% of women, which could be much higher due to underreporting of the disease [24]. In the United States, VVC represents the second most common cause of vaginal infections, affecting 70–75% of women and resulting in an annual treatment cost of at least USD 368 million [25]. In addition, the few points of selective fungal toxicity greatly limit therapeutic options and lead to the topical use of antifungals, contributing to fungal resistance and difficulties in treatment [26,27,28].

Studies evaluating the antifungal effect of araçá fruit extracts are scarce in the literature, making this relevant because of the wealth of bioactive compounds and antioxidant capacity of these fruits. In addition, the search for natural compounds that have pharmacological potential against different types of cancer is of great importance. Therefore, the objective of this work was to evaluate the physicochemical composition, as well as the toxicity of the fruits of yellow and red araçá, and to investigate the potential antiproliferative of cancer cells and the antifungal effects of their extracts.

## 2. Materials and Methods

### 2.1. Chemicals

Chlorogenic acid; Dulbecco’s modified minimum essential medium (DMEM); HEPES buffer; antibiotics penicillin G and streptomycin; MTT (3-(4,5-dimethylthiazolyl)-2,5-diphenyl-2H-tetrazolium bromide); CTT 2,3,5-triphenyltetrazolium chloride—0.5%; and dimethylsulfoxide were purchased from Sigma-Aldrich (St. Louis, MO, USA). Fetal bovine serum was purchased from GIBCO (Carlsbad, CA, USA). All other reagents were of analytical grade.

### 2.2. Plant Material and Preparation of the Extracts

Yellow and red araçá fruits from the cultivars “Ya-cy” and “Irapuã”, respectively, were harvested at the ripe stage at EMBRAPA-CPACT-Pelotas/RS, Brazil, and were washed and subsequently frozen for transport in coolers with ice. After arrival, the fruits remained frozen at −20 °C until they were used for the preparation of the extract and the physicochemical analyses.

To obtain the extracts, fruits were thawed and smashed, and samples of pulp and peels from the fruits were taken. For the extraction of the phenolic compounds, three hundred grams of frozen samples were homogenized with 900 mL of an ethanolic solution (95%) protected from light, using an ultra-turrax mixer (IKA, Campinas, SP, Brazil) for 5 min, then placed on a magnetic stirrer for 30 min. After centrifugation for 5 min at 3000 rpm, the supernatant was collected. The residue was subjected to further extraction as described above, mixing its supernatant with the previous one. The recovered supernatant was evaporated in an evaporator route (39–41 °C, pressure 105 Pa, and 50 rpm). The dry extract obtained was resuspended in milli-q water, being used for the determination of total phenolic compounds [29]. The results of total phenolic compounds were expressed as chlorogenic acid equivalents/mL (CAE/mL) using a standard chlorogenic acid curve.

### 2.3. Physicochemical Analysis of Fruits

The physicochemical analyses of dry matter, ash, and crude protein were carried out according to AOAC methods [30]. Dry matter analysis was carried out using an oven at 105 °C (overnight), and moisture was calculated via difference; ash (mineral residue) analyses were carried out using a muffle furnace (550 °C for 4 h); proteins were analyzed using the Kjeldhal method [30]; lipids were analyzed via the Bligh and Dyer method [31]; pH were analyzed using potentiometry and total acidity [32]. Carbohydrates were calculated using difference as non-nitrogen extracts in the proximate composition of the fruits, i.e., carbohydrates = 100 − (moisture + ash + proteins + lipids).

### 2.4. Toxicological Evaluation of Araçá Extracts

NIH/3T3 cell lines were obtained from the ATCC. The toxicological effects of araçá extracts were evaluated in NIH/3T3 cell lines and were used as a control. Cells were routinely maintained in Dulbecco’s modified minimum essential medium (DMEM) supplemented with 10% fetal bovine serum and 2 g/L HEPES buffer, pH 7.4, containing antibiotics penicillin G 100 U/mL and streptomycin 100 g/mL, maintained at 37 °C and 5% CO_2_ conditions. The cells were plated (2 × 10^5^ cells) in 24-well plates and cultured for 24 h to reach 60–70% of confluence before the treatment with araçá extracts. Araçá extracts were diluted in culture medium to final concentrations of 2.5, 5, 10, 25, 50, 100, 200, 300, 400, and 500 mg CAE/mL, the cell lines were treated with extract concentrations for 24 h. Each concentration group included three or four wells. The control group cell was treated with the only standard medium.

To assess the cytotoxicity of araçá extracts, the MTT (3-(4,5-dimethylthiazolyl)-2,5-diphenyl-2H-tetrazolium bromide) assay was performed. MTT assay is a yellow tetrazolium salt that is reduced to purple formazan crystals, widely used for the assessment of cytotoxicity, cell viability, and proliferation studies in cell biology [33]. After the finish of treatment, cells were incubated with 1 mg/mL MTT for 2 h at 37 °C. Purple crystals were dissolved in dimethylsulfoxide. The absorbance was measured using a spectrophotometric microtiter plate reader (Spectra Max M5; Molecular Devices, Sunnyvale, CA, USA) at 570 nm and 630 nm.

### 2.5. Antiproliferative Activity

Cancer cell lines were obtained from the ATCC. The antiproliferative effect of araçá extracts was evaluated in cancer cell lines glioma C6 (ATCC CCL-107^TM^), colorectal adenocarcinoma HT-29 (ATCC HTB-38^TM^), and prostate carcinoma DU-145 (ATCC HTB-81^TM^). Cells were routinely maintained in Dulbecco’s modified minimum essential medium (DMEM) or RPMI 1640 supplemented with 10% fetal bovine serum and 2 g/L HEPES buffer, pH 7.4, containing antibiotics penicillin G 100 U/mL, and streptomycin 100 g/mL, maintained at 37 °C and 5% CO_2_ conditions. The cells were plated (2 × 10^5^ cells) in 24-well plates and cultured for 24 h to reach 60–70% confluence before treatment with araçá extracts. According to data observed in toxicological evaluation, the final concentrations of araçá extract treatment were 2.5, 5, 10, 25, and 50 mg CAE/mL diluted in a culture medium, and the cell lines were treated for 24 h. Each sample group included three or four wells. The control group for each cell line was treated with the only standard medium. After the treatment period, the MTT assay was performed, according to the method described above.

### 2.6. Antifungal Activity

The antifungal activity was identified using the broth microdilution method, according to the M60 protocol of the Clinical and Laboratory Standards and Institute [34]. Twenty-three clinical isolates from the oral microbiota and three ATCC strains of the genus *Candida* are resistant to fluconazole: twenty-two *C. albicans* and three non-*albicans Candida* (*C. krusei*, *C. glabrata*, and *C. dubliniensis*) were evaluated against red and yellow araçá extracts in the concentration range of 1250 to 2.44 μg CAE/mL in 96-well plates. Fluconazole was used as a control drug at the maximum concentration of 32 mg/mL used in the clinic. The plates were incubated at 35 °C for 24 h and interpretation was performed visually. The minimum inhibitory concentration (MIC) was determined as the lowest concentration evaluated capable of totally inhibiting fungal growth. The minimum fungicide concentration (MFC) was determined using 2,3,5-triphenyltetrazolium chloride—0.5% CTT, after determining the MIC, according to [35] with modifications.

### 2.7. Irritability Test against Vascular Response (HET-CAM)

HET-CAM was performed following the ICCVAM-recommended test method protocol [36]. So, sixty-six fertilized *Gallus domesticus* eggs were incubated in an automatic egg incubator (Pantanal^®^—topchok) at a temperature of 37.8 ± 0.3 °C, a relative humidity of 58 ± 2%, and daily movement. Eggs were divided in 12 groups: negative control (0.6% NaCl), positive control (1 N NaOH), and 10 groups of different concentrations of each araçá extract diluted with insulin (2.44, 4.88, 9.76, 19.53, 39.06, 78.12, 156.25, 312.5, 625, and 1250 µg CAE/mL) (*n* = 3 eggs/group). On the 9th day of incubation, eggs had their chorioallantoic membrane exposed to 300 µL of each extract concentration. The vascularization was evaluated for 5 min for hyperemia, hemorrhage, and/or coagulation. The endpoints were recorded with photographs in 0.5, 2, and 5 min. The numerical time-dependent score values, considering the possible vascular changes and the time-dependent evaluation, were used to generate unique numerical values [37]. Depending on the value obtained, a score below 1 indicated a non-irritating substance, between 1.0 and 4.9 was considered a mild irritant, between 5.0 and 8.9 characterized moderate irritation, and above 9.0 represented severe irritability.

This model does not conflict with ethical and legal obligations since the central nervous system of the chick is too incomplete to suffer or perceive pain. HET-CAM has regulatory acceptance in several countries [38].

### 2.8. Statistical Analysis

Data were reported as mean ± standard deviation (mean ± SD). The results of the physicochemical data were analyzed using the paired *t*-test using significance (*p* < 0.05). The data on antiproliferative activity were submitted to one-way ANOVA followed by Tukey’s test (*p* < 0.05). All analyses and graphs were performed using the GraphPad PRISM (version 7.0 for Windows; Graph Pad Software).

## 3. Results and Discussion

Araçá is a plant native to southern Brazil, but it is a little-consumed fruit due to the population’s unfamiliarity with it. Araçazeiro leaves are widely used in traditional medicine in the form of teas and stews, and there are already several studies evaluating their pharmacological properties [1,2,3,5,9,10,13,14,16,17,18]. On the other hand, studies evaluating the composition and biological effects of araçá fruits are still scarce; therefore, it is essential to undertake research to stimulate the consumption of araçá fruits and promote their potential and promising pharmacological applications.

In this work, we observed that although the yellow and red araçá fruits are the same species, they present marked differences in physicochemical composition (Table 1). In general, we observed that the araçá is a fruit rich in nutrients and antioxidant compounds, especially dietary fiber. Our results show that red araçá has a significantly higher content of dry matter, carbohydrates, total and insoluble dietary fiber, protein, caloric value, pH, and total soluble solids, when compared to yellow araçá. However, yellow araçá showed a greater amount of minerals (ash), water, and soluble fiber (Table 1). We emphasize that although they are the same species, the fruits present marked differences in chemical composition and can thus be used for different nutritional goals. A study that reviewed the composition and bioactive compounds of araçá demonstrated that in 100 g of araçá fresh fruit there is 81.73–84.9 g of water, 0.75–1.03 g of protein, 0.63–1.50 g of minerals, 4.32–10.01 g of carbohydrate, 0.42–0.55 g of lipid, 3.87–6.14 g of fiber, and 26.8 kcal of energy [4], which is in agreement with results observed in our study. Therefore, we can demonstrate that red araçá fruits are an excellent source of dietary fiber, especially concerning insoluble fiber, and that both araçá fruits are acid (pH around 3.6), with low caloric content and rich in total phenolic compounds, which could enhance their consumption by the population.

The phytochemical evaluation demonstrated that the red and yellow araçá had similar amounts of total phenolic compounds, around 5209–5517 µg CAE/mL, respectively (Table 1). A similar profile of araçá phenolic compounds was observed in previous work performed by our laboratory, where yellow araçá showed a larger amount of total phenols, followed by flavonoid content, than red araçá [39]. On the other hand, a different profile was found by Biegelmeyer and colleagues when comparing the red and yellow araçá and demonstrated that red araçá total content of phenolic compounds, expressed in fresh weight, was more abundant than yellow (501.33–292.03 mg of GAE/100 g, respectively) [7]. Other studies demonstrated that the total content of phenolic compounds did not differ between yellow and red genotypes (5372 mg and 5638 mg of gallic acid equivalent (GAE)/100 g and 603 mg and 606 mg of CAE/100 g, respectively) [40,41], which are very similar to those observed by our group. Although we did not observe significant differences in the amount of total phenolic compounds between the yellow and red fruits, previous work by our group on the characterization of yellow and red araçá extracts showed differences in the composition of major compounds. Both fruits showed remarkably similar amounts of hydroxybenzoate derivatives and flavonols; however, the yellow araçá extract showed a significant number of hydroxycinnamate derivatives while the red araçá had a higher anthocyanin content [39]. These characteristics suggest the pharmacologically beneficial effects of araçá fruits.

To assess the possible cytotoxicity of yellow and red araçá fruit extracts, the cell viability test using the NIH/3T3 fibroblast cell line was performed. We observed a significant reduction in cell viability from the concentration of 100 µg CAE/mL for both yellow and red araçá extracts in 24 h of treatment (Figure 1). Based on these results, the inhibitory concentration of 50% (IC_50_) was obtained at 73.60 µg CAE/mL for yellow and 83.38 µg CAE/mL for red araçá extracts. A study demonstrated that aqueous and acetonic extracts of red and yellow araçá did not affect the survival of control cells (NIH/3T3 rat embryonic fibroblasts) at concentrations of 40, 60, and 80 µg/mL. However, the same study showed that araçá extracts reduced the proliferation of cancer cell lines MCF-7 (mammary cancer cells) and Caco-2 (colon cancer cells) [20]. The reduction in viability observed in our work may be related to the concentration of extract used in the cells, since different studies have used concentrations expressed in ug/mL of extract and we used concentrations expressed in phenolic compound content. We always choose to express the concentrations used as phenolic compound content to avoid variations due to harvest, post-harvest, and fruit processing factors. This ensures that we always have the same number of bioactive compounds acting in the sample.

Cancer is one of the main causes of morbidity and mortality worldwide [42,43,44]. To evaluate the potential effect of red and yellow araçá ethanolic extract in cancer cell lines, we used cell lines of *Rattus norvergicus* brain glioma (C6), human prostate carcinoma (DU-145), and human colorectal adenocarcinoma (HT-29) (Figure 2). Based on the IC_50_ observed in this study, concentrations less than 50 µg EAC/mL of the extracts were chosen to evaluate antiproliferative effects in cancer strains. Among all the concentrations tested, only the extract of red araçá showed a significant reduction in cell viability at a concentration of 50 µg EAC/mL in the HT-29 strain (Figure 2F). No reduction in cell viability was observed in the other cell lines evaluated (Figure 2). Chaves and colleagues (2018), evaluated the biological proprieties of Brazil berries, including red and yellow araçá methanolic extract, and the cytotoxic effect on the cancer cells lines A549 (non-small cell lung cancer cells), RD (rhabdomyosarcoma), and DU145 (prostate carcinoma); however, none of the extracts were able to reduce cell viability in all tested cancer cell lines [45]. We suggest that the effect observed in the red araçá extract and not in the yellow one was due to the presence of anthocyanins content in their composition, as was observed in the composition of the majority phenolic compounds of red araçá [39]. Furthermore, other studies using fruit extracts rich in anthocyanins, such as grapes (*Vitis vinifera*), blueberries (*Vaccinium myrtillus* L.), and chokeberry (*Aronia meloncarpa* E.), were able to inhibit the growth of HT-29 cells [46]. Therefore, we believe that red araçá extract has an antiproliferative effect on cancerous cells regardless of its toxicity.

Is important to emphasize that one of the main risk factors for cancer is the reduced consumption of fruits and vegetables by the population. Among the benefits of consuming fruits and vegetables, we highlight their antioxidant effect, which works by stimulating the activity and expression of antioxidant enzymes, providing essential micronutrients and strengthening the immune system. In addition, dietary fibers, present in large quantities in araçá fruits, are able to increase fecal volume and, consequently, intestinal transit, which reduces the absorption of potentially carcinogenic agents [47]. Thus, the Ministry of Health (MS) of Brazil recommends a minimum consumption of fruits and vegetables of 400 g/day [48] as a preventive measure for the appearance of some types of cancers, such as colorectal cancer. Thus, we suggest that the consumption of araçá fruits could be expanded and used as a source of fibers and bioactive compounds beneficial to health.

The search for new drugs or substances with antimicrobial or antifungal potential has grown considerably over the years, due to the indiscriminate use of antimicrobials, both by preventive medicine and agriculture for pest control, thus considerably increasing microbial resistance [26]. Thus, the use of extracts and natural essential oils has been an alternative to the use of antimicrobials, and subsequently, seeking new alternatives that have equivalents results in reference drugs and reduced side effects [14,17,26]. Here, we evaluated the antifungal effects of yellow and red araçá extracts on *Candida* spp. (Table 2), and for the first time in the literature, we reported a pronounced antifungal effect of araçá fruit extracts on *Candida albicans*.

The antifungal effect of yellow and red araçá extracts was evaluated in 22 clinical isolates of *Candida albicans* and 4 clinical isolates of non-*albicans Candida* in comparison to the reference drug fluconazole (Table 2). We observed that the reference drug, fluconazole, did not have an antifungal effect in any of the *Candida* spp. tested when used at the maximum clinic concentration of 32 mg/mL, demonstrating that all *Candida* spp. used were resistant to fluconazole. In general, the results showed that non-*albicans Candida* proved to be more resistant to both araçá extracts based on MIC and MFC concentrations (Table 2).

Moreover, we observed that araçá extracts, regardless of the variety, showed significant antifungal activity in the *Candida albicans* isolates. Among the *Candida* spp. evaluated in this study, only the isolates *C. albicans* 2836 F, *Candida dubliniensis* ATCC 7957, and *C. glabrata* ATCC 2001 were not inhibited by both araçá extracts. However, the red araçá extract showed a more effective inhibition, as it presented lower values of the geometric mean (GM) (MIC 40.6 µg EAC/mL and MFC 41.7 µg EAC/mL) than those of yellow araçá (MIC 44.9 µg EAC/mL and MFC 48.7 µg EAC/mL). The antifungal effect at a concentration less than or equal to 39.06 µg EAC/mL was observed in 18 of the 22 isolates of *C. albicans* evaluated against the extract of red araçá and 19 of the 22 isolates of *C. albicans* in relation to yellow araçá. However, at these same concentrations, only the extract of red araçá showed activity for two of the four non-*albicans Candida* isolates. These same isolates showed inhibition by the extract of yellow araçá but at a higher concentration (78.12 µg EAC/mL) (Table 2).

Other authors have already reported that the araçá leaves have antimicrobial [13,16] and antifungal effects [10,13,14,17]. Scur and colleagues [14] observed that the araçá leaves ethanolic extract showed the highest antimicrobial activity against the microorganisms tested when compared to the aqueous extract. In the ethanolic extract, the MIC ranged from 0.78 to 25 mg/mL for Gram-negative bacteria, from 0.78 to 3.125 mg/mL for Gram-positive bacteria, and exhibited a MIC of 3.125 mg/mL for yeast *C. albicans* ATCC [14]. In our work, the extracts from araçá fruits were able to inhibit 21 of the 22 isolates of *C. albicans* with 156 µg/mL, that is, a much lower concentration than that observed for the leaves. We found only one study that evaluated the antimicrobial effect of the araçá fruit extract. In this work, the water and acetone extracts of yellow and red araçá fruits were evaluated. The antimicrobial activity against *S. enteritidis* was evaluated by looking at the formation of the inhibition halo and determining the minimal inhibitory concentration (MIC) of the extracts. All araçá extracts showed antimicrobial activity and the MIC was 5%, except for the water extract of red araçá (16%) [20].

The antimicrobial activity of plants is mainly related with the presence of phenolic compounds in leaves and fruits [49,50]. Araçá contains flavonoids (i.e., kaempferol and quercetin) and anthocyanidins (cyanidin) that are well recognized as antimicrobial agents. The mode of action of these compounds is related to their reaction with microbial cellular membrane by inactivating essential enzymes or forming complexes with metallic ions, limiting their accessibility to the microbial metabolism [20]. It has been shown that extracts from fruits that are rich in secondary metabolites usually prevent bacterial cell proliferation. Phenolic compounds could act as a destabilizing bacterial cell membrane that is primarily responsible for the respiration of these microorganisms. In yeast, phenolics could act as metal chelators or scavenging free radicals, which are otherwise harmful to the cell. In this case, the cell membrane is not harmed and, consequently, mitochondria, which are vital for respiration, are not affected by the action of phenolic compounds [20,51].

The mechanisms of action of herbal medicine in the treatment of *Candida* spp. are still poorly studied. Several mechanisms of action have been proposed from the rupture of the cellular membranes, which seems to interrupt the cell cycle through the synthesis of proteins and the alteration of the yeast DNA [52]. We believe that the araçá extracts used in our study had expressive antifungal activity in *Candida* spp. due to the richness of phenolic compounds in the composition.

Since the antifungal effect under *Candida* spp. was observed, the irritant potential of araçá extracts was evaluated to examine the development of a formulation for the treatment of candida infections. For this, we used an irritability test in the chorioallantoic membrane (HET-CAM) [37]. The use of the HET-CAM test to evaluate the irritability of plant extracts as sources for substances through topical application was been already described [53]. We observed that all tested concentrations in HET-CAM for both yellow and red araçá extracts had no negative effect on the vascularization of the chorioallantoic membrane, with all considered non-irritating (Figure 3). In addition to safety for ocular applications, evaluation via HET-CAM is also related to the irritant potential in other epitheliums, which can be correlated to the mucous/substrates, such as the vaginal epithelium, demonstrating that its application through different routes of administration are favored [54]. The positive control (NaOH 0.1 N) reached a score of 9, considered a severe irritant, and was different from the negative control that presented a score of 0, being non-irritating (Figure 3), thus validating these assessments, according to what is recommended by the Interagency Coordinating Committee on the Validation of Alternative Methods [37].

## 4. Conclusions

Yellow and red araçá are small fruits common in southern Brazil and are rich in nutrients, mainly phenolic compounds; vitamin C; and total dietary fiber. Here, we observed that the extract of red araçá fruit showed a cytotoxic effect against colorectal cancer, which is very promising and needs further research, since this type of cancer is one of the most common among men and women. Furthermore, we highlight the antifungal effect in isolates of the genus *Candida* spp. that are resistant to fluconazole, where both araçá extracts were effective in low concentrations, with emphasis on the red araçá extract. In addition, both araçá extracts did not show irritant effects, suggesting suitability for topical application. Thus, we conclude that the extracts of the yellow and red araçá fruits have promising biological effects. In the future, we aim to explore the mechanisms of action and the pharmacological potential of the extracts as antifungal and antiproliferative agents. Furthermore, we intend to develop and test topical pharmaceutical methods involving the extract of red araçá fruits for the treatment of vaginal candidiasis.

## Figures and Tables

**Figure 1 foods-12-04307-f001:**
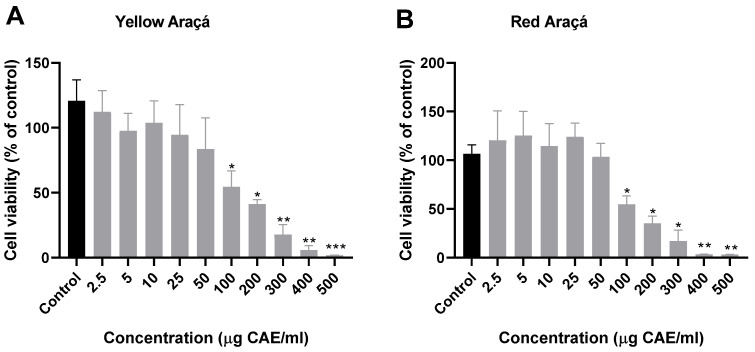
Toxicological evaluation of araçá fruit extracts using NIH/3T3 cells using MTT assay. (**A**) yellow araçá extract and (**B**) red araçá extract. Data correspond to the average of three independent experiments ± SD. * *p* < 0.05; ** *p* < 0.01; *** *p* < 0.001 indicate significant differences between treated and control cell lines (ANOVA with Tukey post-test).

**Figure 2 foods-12-04307-f002:**
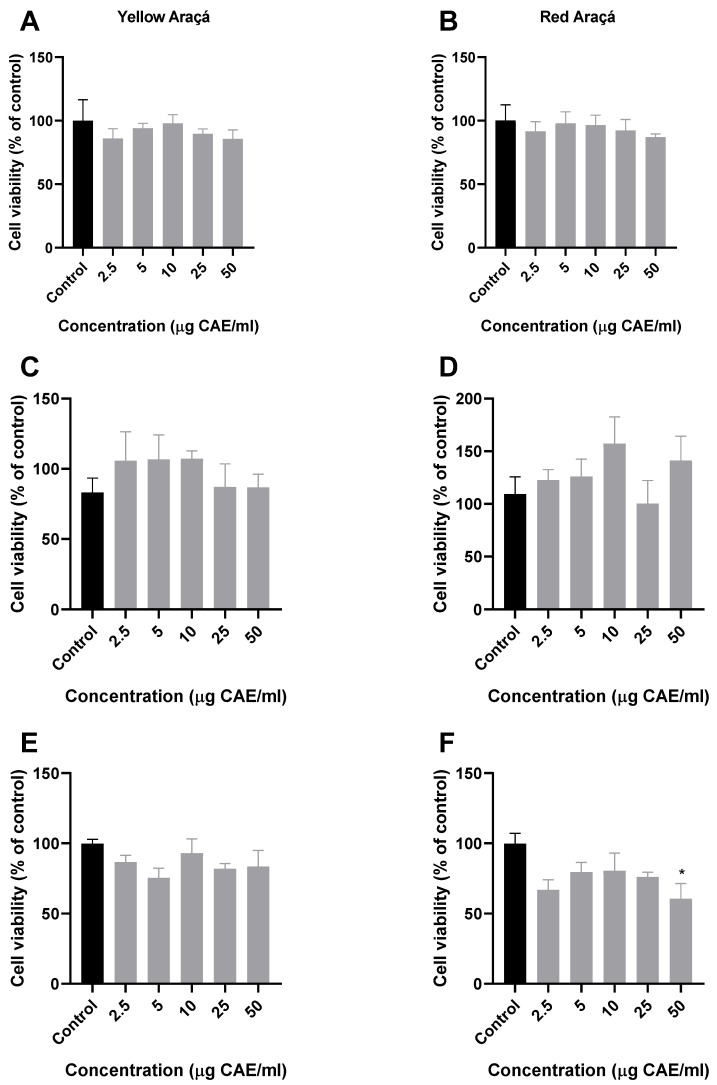
Effect of araçá fruit extracts on viability of cancer cell lines using MTT assay. (**A**) C6 glioma cell line, (**C**) DU145 prostate cancer cell line, and (**E**) HT-29 colorectal cancer cell line exposed to yellow araçá extract (left) and (**B**) C6 glioma cell line, (**D**) DU145 prostate cancer cell line, and (**F**) HT-29 colorectal cancer cell line exposed to red araçá extract (right). Data correspond to the average of three independent experiments ± SD. * *p* < 0.05 indicates significant differences between treated and control cell lines (ANOVA with Tukey’s post hoc test).

**Figure 3 foods-12-04307-f003:**
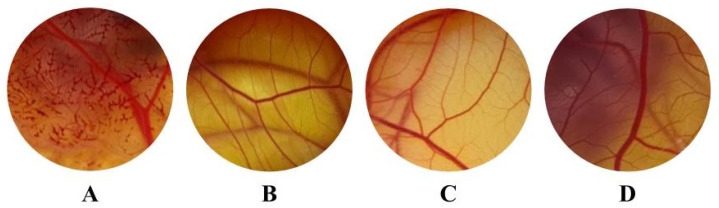
Photograph at 5 min of the chorioallantoic membrane of the positive control (**A**), negative control (**B**), and 1250 µg CAE/mL of yellow (**C**) and red (**D**) araçá.

**Table 1 foods-12-04307-t001:** Physicochemical composition of yellow and red araçá fruits.

	Yellow Araçá	Red Araçá
Dry matter	18.14 ± 0.16 ^b^	20.75 ± 0.09 ^a^
Moisture	81.86 ± 0.16 ^a^	79.25 ± 0.09 ^b^
Ash (mineral residue)	0.90 ± 0.01 ^a^	0.79 ± 0.01 ^b^
Carbohydrates	16.26 ± 0.23 ^b^	18.56 ± 0.12 ^a^
Total dietary fiber	5.36 ± 1.51 ^b^	7.80 ± 1.02 ^a^
Soluble fiber	2.43 ± 1.33 ^a^	0.66 ± 0.45 ^b^
Insoluble fiber	2.93 ± 0.18 ^b^	7.48 ± 1.96 ^a^
Crude protein	0.74 ± 0.02 ^b^	0.92 ± 0.04 ^a^
Total lipids	0.67 ± 0.12	0.56 ± 0.14
Total energy value	69.11 ± 1.87 ^b^	74.17 ± 0.75 ^a^
pH	3.56 ± 0.00 ^b^	3.61 ± 0.01 ^a^
Total acidity (100 g of citric acid)	24.51 ± 1.03	23.02 ± 0.22
Total soluble solids (°Brix)	10.01 ± 0.11 ^b^	13.50 ± 0.14 ^a^
Phenolic compounds (µg chlorogenic acid equivalent/mL)	5517.87 ± 482.98	5209.61 ± 302.09

Values expressed in % (g/100 g fresh weight) and total energy value (Kcal/100 g fresh weight) of fruits in mean ± SD. The means followed by different letters show significant differences with *p* < 0.05 (*n* = 3).

**Table 2 foods-12-04307-t002:** Number of isolates with minimum inhibitory concentration (MIC) and minimum fungicide concentration (MFC) of yellow and red araçá fruit extracts.

Isolated	Yellow Araçá ^1^		Red Araçá ^1^		Fluconazole ^2^	
	MIC	MFC	MIC	MFC	MIC	MFC
*Candida albicans* 15 A	9.76	9.76	9.76	9.76	>320	>320
*Candida albicans* 10 A	39.06	39.06	156.25	156.25	>320	>320
*Candida albicans* 10 G	19.53	19.53	19.53	19.53	>320	>320
*Candida albicans* 14 A	39.06	39.06	19.53	19.53	>320	>320
*Candida albicans* ATCC 10231	78.12	78.12	39.06	39.06	>320	>320
*Candida albicans* 24 C	9.76	9.76	78.12	78.12	>320	>320
*Candida albicans* 26 A (n 3)	39.06	39.06	19.53	19.53	>320	>320
*Candida albicans* 2836 F	>1250	>1250	>1250	>1250	>320	>320
*Candida albicans* 3 H	39.06	39.06	19.53	19.53	>320	>320
*Candida albicans* 3 Q	39.06	39.06	19.53	19.53	>320	>320
*Candida albicans* 49 F	9.76	9.76	39.06	39.06	>320	>320
*Candida albicans* 55 A	19.53	19.53	19.53	19.53	>320	>320
*Candida albicans* 6 E	19.53	19.53	19.53	19.53	>320	>320
*Candida albicans* 6231	39.06	39.06	39.06	39.06	>320	>320
*Candida albicans* 66 A	9.76	9.76	19.53	19.53	>320	>320
*Candida albicans* 69 A	39.06	39.06	19.53	19.53	>320	>320
*Candida albicans* 69 F	78.12	78.12	78.12	78.12	>320	>320
*Candida albicans* 76 E	39.06	39.06	9.76	9.76	>320	>320
*Candida albicans* 8 F	39.06	78.12	39.06	78.12	>320	>320
*Candida albicans* 91 A	39.06	39.06	19.53	19.53	>320	>320
*Candida albicans* 49 F	19.53	19.53	19.53	19.53	>320	>320
*Candida albicans* 55 A	19.53	19.53	19.53	19.53	>320	>320
*Candida dubliniensis* ATCC 7957	1250.00	1250.00	625.00	625.00	>320	>320
*Candida glabrata* 61 B	78.12	78.12	39.06	39.06	>320	>320
*Candida glabrata* ATCC 2001	>1250	>1250	>1250	>1250	>320	>320
*Candida krusei* 85 A	78.12	78.12	39.06	39.06	>320	>320

Results expressed in MIC and MFC (*n* = 3). ^1^ Concentration of the extract in µg of chlorogenic acid equivalent/mL. ^2^ Concentration of fluconazole expressed in (µg/mL). Fluconazole was used as a control drug in the maximum concentration of 32 mg/mL used in the clinic. *C. albicans* (*n* = 22); non-*albicans Candida* (*n* = 4).

## Data Availability

The data presented in this study are available on request from the corresponding author.

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
