# Peer review of "Antiproliferative Cancer Cell and Fungicidal Effects of Yellow and Red Araçá (Psidium cattleianum Sabine) Fruit Extract"

_foods, 2023, doi:10.3390/foods12234307_

Round 1
Reviewer 1 Report
Comments and Suggestions for Authors
This study aimed to evaluate the physicochemical characteristics and in vitro toxicity, antiproliferative and antifungal effects of araçá extracts from two yellow and red varieties. The novelty of the research should be better substantiated. There are many typographical and linguistic errors, authors should correct those carefully. For instance, in the title, the name of the specie should be in italic, and should corrected in line 26. A section dedicated to “Reagents and standards” should be included in the experimental section. Moreover, the discussion of the results and main conclusions of the manuscript should be strongly improved. References should be corrected, there are a lot of typographical and linguistic errors, and the name of some species should be written in italic.
The present status of the manuscript does not meet the standard of the journal. For the improvement, please see some examples of suggestions/comments:
ABSTRACT:
Line 19: authors use “…/ml”, “.../mL” (line 80) or “…L-1” (line 91) >> please uniformize according to journal´s instructions throughout the manuscript.
Line 21: should be “showed no ….”.
Line 22: should be “promising biological and pharmacological effects...”
Line 26: should be “araçá”
Line 26: should be “Psidium cattleianum Sabine”.
Line 27: remove space after “between”
Line 33-34: please, give some examples of these classes of compounds.
Lines 33-44: this section is very confused. The findings related to fruits should be distinguished from those for leaves.
Line 39: authors stated that “However, it should be considered that most of these studies are concerned with the use of leaves and few studies evaluate the fruits” but the references used are most related to fruits.
Line 62: the interest of this research topic should be better clarified.
MATERIAL AND METHODS
Line 63: The is no information about the standards and reagents used in the experiments. Authors should include this information through the manuscript or in a dedicated subsection.
Line 68: Any washing procedure? It seems extremely important!
Line 73: it is missing the information about the equipment used
Line 82: Was the dry matter calculated from the humidity value? If so, it should be not considered an analysis
Line 84: it is missing the reference for the Bligh and Dyer method
Line 84: it shoud be “Bligh and Dyer method”
Line 85: “Carbohydrates were calculated by difference as non-nitrogen extracts in the proximate composition of the fruits. >> please clarify
Line 106: include space before “nm”
Line 130: introduce the abbreviation MIC
Lines 155-158: “P” or “p” please uniformize according to journal´s instructions throughout the manuscript.
RESULTS AND DISCUSSION
Table 1 refers to nutritional and physicochemical parameters. Thus, the title and the discussion section presented in lines 167-183 should be modified accordingly.
Lines 185-187: needs correction. >> 100 g. wet basis (??), Use the abbreviation already presented SD for standard deviation. Explain “T” test.
Lines 201-205: include the mean of contents reported by literature.
Lines 216-219: sentence needs edition.
Line 221: “3T3 cells” ?? >> the correct is “NIH/3T3 cell lines”
Line 226: aracá??
Line 228: it should be IC50
Line 235-2450: wrong format
Line 270: Authors stated that “for the first time in the literature, we reported a pronounced
antifungal effect of araçá fruit extracts on Candida albicans.”. This topic was already investigated by other authors (https://doi.org/10.5897/JMPR2019.6790, “Antimicrobial activity and acetilcolinesterase inhibition of oils and Amazon fruit extracts”).
Lines 247-251. It is missing the information about the statistical method used. The horizontal axis needs edition>> it should be “2.5” instead of “2,5”.
Table 2: Could authors give any explanation to the high values observed for Candida dubliniensis 7957 and Candida glabrata2001 fungus?
Line 253 : “Among the benefits of the consumption of fruits and vegetables, are antioxidant effects,…” >> edit English.
Comments on the Quality of English LanguageThere are many typographical and linguistic errors, authors should correct those carefully. References should be corrected, there are a lot of typographical and linguistic errors, and the name of some species should be written in italic.
Author Response
Response to review
First, we would like to thank you for your contributions to the work; for sure, this manuscript has improved a lot due to your contributions.
Minor concerns:
For better presentation, as the list is long, I will comment only those that have generated some questioning between the authors. All others have been modified or corrected as requested.
All modifications made to the text were marked throughout the manuscript.
Reviewers' comments:
Reviewer #1:
Major comments:
This study aimed to evaluate the physicochemical characteristics and in vitro toxicity, antiproliferative and antifungal effects of araçá extracts from two yellow and red varieties. The novelty of the research should be better substantiated.
We appreciate the suggestion, and we modified the introduction.
There are many typographical and linguistic errors, authors should correct those carefully. For instance, in the title, the name of the specie should be in italic, and should corrected in line 26.
The text has been extensively revised and we believe that all overstatements have been modified.
A section dedicated to “Reagents and standards” should be included in the experimental section.
Thanks for the comment. We inserted a Chemical section.
Moreover, the discussion of the results and main conclusions of the manuscript should be strongly improved.
Thanks for the comment. We have revised and improved the discussion and conclusion.
References should be corrected, there are a lot of typographical and linguistic errors, and the name of some species should be written in italic.
The text has been extensively revised and we believe that all overstatements have been modified.
The present status of the manuscript does not meet the standard of the journal. For the improvement, please see some examples of suggestions/comments:
The manuscript was extensively revised and sent for correction by a native English speaker.
ABSTRACT:
Line 19: authors use “…/ml”, “.../mL” (line 80) or “…L-1” (line 91) >> please uniformize according to journal´s instructions throughout the manuscript.
The text has been extensively revised and we believe that all overstatements have been modified.
Line 21: should be “showed no ….”.
The text has been extensively revised and we believe that all overstatements have been modified.
Line 22: should be “promising biological and pharmacological effects...”
The text has been extensively revised and we believe that all overstatements have been modified.
Line 26: should be “araçá”
The text has been extensively revised and we believe that all overstatements have been modified.
Line 26: should be “Psidium cattleianum Sabine”.
The text has been extensively revised and we believe that all overstatements have been modified.
Line 27: remove space after “between”
The text has been extensively revised and we believe that all overstatements have been modified.
Line 33-34: please, give some examples of these classes of compounds.
Thanks for the comment. We have revised and improved this paragraph.
Lines 33-44: this section is very confused. The findings related to fruits should be distinguished from those for leaves.
Line 39: authors stated that “However, it should be considered that most of these studies are concerned with the use of leaves and few studies evaluate the fruits” but the references used are most related to fruits.
Thanks for the comment. We have revised and improved this paragraph.
Line 62: the interest of this research topic should be better clarified.
Thanks for the comment. We have revised and improved this paragraph.
MATERIAL AND METHODS
Line 63: The is no information about the standards and reagents used in the experiments. Authors should include this information through the manuscript or in a dedicated subsection.
Thanks for the comment. We inserted a Chemical section.
Line 68: Any washing procedure? It seems extremely important!
Thanks for the comment. We inserted this information.
Line 73: it is missing the information about the equipment used
Thanks for the comment. We inserted this information.
Line 82: Was the dry matter calculated from the humidity value? If so, it should be not considered an analysis
Thanks for the comment. When we evaluate the moisture of a food, we measure the dry matter. The humidity value is calculated by difference. We chose to leave both values in the table to make it more complete and because it is the standard found in food composition tables.
Line 84: it is missing the reference for the Bligh and Dyer method
Thanks for the comment. The reference is inserted. [31]
Line 84: it shoud be “Bligh and Dyer method”
Thanks for the comment. Corrected in the text.
Line 85: “Carbohydrates were calculated by difference as non-nitrogen extracts in the proximate composition of the fruits. >> please clarify
Thanks for the comment. The standard formula used for carbohydrate analysis by difference has been inserted.
Line 106: include space before “nm”
The text has been extensively revised and we believe that all overstatements have been modified.
Line 130: introduce the abbreviation MIC
Thanks for the comment. Corrected in the text.
Lines 155-158: “P” or “p” please uniformize according to journal´s instructions throughout the manuscript.
Thanks for the comment. Corrected in the text.
RESULTS AND DISCUSSION
Table 1 refers to nutritional and physicochemical parameters. Thus, the title and the discussion section presented in lines 167-183 should be modified accordingly.
Thanks for the comment. Corrected in the text.
Lines 185-187: needs correction. >> 100 g. wet basis (??), Use the abbreviation already presented SD for standard deviation. Explain “T” test.
Thanks for the comment. wet basis means that the results are expressed on fresh fruit, considering the moisture content. This is a common term in food analysis tables, but we can replace it with the term fresh fruit.
Lines 201-205: include the mean of contents reported by literature.
Thanks for the comment. Inserted in the text.
Lines 216-219: sentence needs edition.
Thanks for the comment. Corrected in the text.
Line 221: “3T3 cells” ?? >> the correct is “NIH/3T3 cell lines”
Thanks for the comment. Corrected in the text.
Line 226: aracá??
Thanks for the comment. Corrected in the text.
Line 228: it should be IC50
Thanks for the comment. Corrected in the text.
Line 235-2450: wrong format
Thanks for the comment. Corrected in the text.
Line 270: Authors stated that “for the first time in the literature, we reported a pronounced antifungal effect of araçá fruit extracts on Candida albicans.”. This topic was already investigated by other authors (https://doi.org/10.5897/JMPR2019.6790, “Antimicrobial activity and acetilcolinesterase inhibition of oils and Amazon fruit extracts”).
This will be the first work to date demonstrating the effect of extracts from araçá "fruits" on Candida. This work cited in the literature used essential oil. There are few studies showing the effect on fungi of leaf extract, essential oil and volatile compounds from fruits.
Lines 247-251. It is missing the information about the statistical method used. The horizontal axis needs edition>> it should be “2.5” instead of “2,5”.
Thanks for the comment. Corrected in the text.
Table 2: Could authors give any explanation to the high values observed for Candida dubliniensis 7957 and Candida glabrata2001 fungus?
We observed that the non-albicans Candida tested showed greater resistance to both the reference medicine and the extracts. They are already classified as the most resistant fungi. However, the most important thing is that we observed effects of the extracts on isolates that were already resistant to fluconazole, a medicine used in the clinic.
Line 253 : “Among the benefits of the consumption of fruits and vegetables, are antioxidant effects,…” >> edit English.
Thanks for the comment. Corrected in the text.

Reviewer 2 Report
Comments and Suggestions for Authors
Antiproliferative and fungicidal effects of yellow and red Aracá fruit extract (Psidium cattleianum Sabine) is the title of the manuscript. This study aimed to investigate the potential health benefits of aracá fruit. An assessment was conducted on the physicochemical properties and in vitro toxicological impacts of red and yellow aracá fruit. The present study assessed the toxicity of aracá extracts on 3T3 cell lines, as well as their antiproliferative and antifungal effects on cancer cell lines (C6, HT-29, and DU149). Using the HET-CAM assay, the irritant potential of aracá extracts was evaluated. The manuscript is organized and well-written. Nevertheless, a few concerns must be addressed.
Please conduct a comparative analysis of the chemical profiles of red and yellow aracá fruit; A minimum requirement is the HPLC method.
It is recommended that the anti-cancer properties of the plant be disclosed during the introductory and discussion sections.
Please provide justification the use of CAE/ml as the unit of concentration; failing to do so would make it difficult to compare the results that were achieved with those that were published in earlier publications. It should typically display gallic acid equivalent per milliliter or milligrams per gram.
What does GL represent in line 73?
Who identified the plant species, and where was the specimen deposited?
Kindly provide information on the origin of the clinically isolated Candida species.
It would be helpful if you could explain in the discussion why the authors chose fluconazole as a positive control for Candida strains that are resistant to fluconazole. Is there any medication available that can treat this particular strain of Candida?
Author Response
Response to review
First, we would like to thank you for your contributions to the work; for sure, this manuscript has improved a lot due to your contributions.
Minor concerns:
For better presentation, as the list is long, I will comment only those that have generated some questioning between the authors. All others have been modified or corrected as requested.
All modifications made to the text were marked throughout the manuscript.
Reviewers' comments:
Reviewer #2:
Antiproliferative and fungicidal effects of yellow and red Aracá fruit extract (Psidium cattleianum Sabine) is the title of the manuscript. This study aimed to investigate the potential health benefits of aracá fruit. An assessment was conducted on the physicochemical properties and in vitro toxicological impacts of red and yellow aracá fruit. The present study assessed the toxicity of aracá extracts on 3T3 cell lines, as well as their antiproliferative and antifungal effects on cancer cell lines (C6, HT-29, and DU149). Using the HET-CAM assay, the irritant potential of aracá extracts was evaluated. The manuscript is organized and well-written. Nevertheless, a few concerns must be addressed.
Please conduct a comparative analysis of the chemical profiles of red and yellow aracá fruit; A minimum requirement is the HPLC method.
Thanks for the comment. A comparative analysis of the extracts using HPLC/DAD was carried out in the previous study and is mentioned in the work; reference 39.
It is recommended that the anti-cancer properties of the plant be disclosed during the introductory and discussion sections.
Thanks for the comment. The anti-cancer properties of the fruits are discussed; see lines 252 to 271.
Please provide justification the use of CAE/ml as the unit of concentration; failing to do so would make it difficult to compare the results that were achieved with those that were published in earlier publications. It should typically display gallic acid equivalent per milliliter or milligrams per gram.
The content of phenolic compounds is typically expressed in gallic acid equivalents or chlorogenic acid equivalents. In the case of araçá fruits, we chose to express it in chlorogenic acid equivalents since we found chlorogenic acid in the extracts in the HPLC/DAD analyzes of previous studies. Therefore, we chose to express it this way.
We chose to express the results in EAC/ml concentration since this minimizes variations in compounds in the fruits between different harvests, post-harvest factors, among others. We always have the same number of phenolic compounds acting in the tests.
What does GL represent in line 73?
Thanks for the comment. Corrected in the text.
Who identified the plant species, and where was the specimen deposited?
The plans are deposited and supplied by Embrapa; a Brazilian government company that works to promote expansion and studies with plants native to the region and supply seedlings and seeds.
Kindly provide information on the origin of the clinically isolated Candida species.
Thanks for the comment. The information was inserted into the text.
It would be helpful if you could explain in the discussion why the authors chose fluconazole as a positive control for Candida strains that are resistant to fluconazole. Is there any medication available that can treat this particular strain of Candida?
Thanks for the comment. Our objective was to evaluate only candida isolates and strains that were resistant to fluconazole. That's why we used fluconazole to prove this resistance. Fluconazole is the antifungal medication most used by the population and in clinics to treat fungal infections. Our goal here was to show that extracts can have a greater effect than an antifungal already used in the clinic. We are already testing and comparing the effect of the extracts with other antifungals such as clotrimazole and others.

Reviewer 3 Report
Comments and Suggestions for Authors
The introduction part could be more detailed strictly about yellow and red araçá fruits.
The Material and methods part is correctly detailed, the results are processed statistically accordingly, they are appreciated comparatively to another research on the same topic.
However, I recommend correcting the following mistakes or clarifying some problems.
- The introduction part should be improved. For example, do climatic and soil conditions influence the content of beneficial compounds of acacia fruits? Enter the references.
- In a natural order, I recommend reversing subchapter 2.2 with 2.1
L 185 Footnote table 1, the values must be expressed g/100 gr FW (fresh weight)
Figure 1. L 223 To correct the P values for the confidence interval
Figure 1. A- How do you explain values over 100% for control in the case of cell viability under treatment with yellow acacia?
Fig 2.C – How do you explain the control value of less than 100% ??
L 247-250 Correct and add information regarding 2B, D and F.
L 250. Correct the value p<0.05
L 276. n represents the number of repetitions used to determine MIC/ MFC for each of the Candida isolates. Correct as required.
Table 2. The maximum concentration of fluconazole taken in the study is 32mg/ml, according to materials and methods. To be corrected in Table 2 and footnote.

The manuscript is well written, with few grammatical mistakes that can be fixed by consulting a native English speaker.
Author Response
Response to review
First, we would like to thank you for your contributions to the work; for sure, this manuscript has improved a lot due to your contributions.
Minor concerns:
For better presentation, as the list is long, I will comment only those that have generated some questioning between the authors. All others have been modified or corrected as requested.
All modifications made to the text were marked throughout the manuscript.
Reviewer #3:
The introduction part could be more detailed strictly about yellow and red araçá fruits.
The Material and methods part is correctly detailed, the results are processed statistically accordingly, they are appreciated comparatively to another research on the same topic.
However, I recommend correcting the following mistakes or clarifying some problems.
- The introduction part should be improved. For example, do climatic and soil conditions influence the content of beneficial compounds of acacia fruits? Enter the references.
Thanks for the comment. It is already well documented in the literature that several factors influence the amount and types of bioactive compounds in plants. To minimize these variations, we use extracts in concentrations equivalent to total phenolic compounds. Thus, we minimize variations by always adjusting the concentration used to the amount of bioactive compounds that the extract has.
- In a natural order, I recommend reversing subchapter 2.2 with 2.1
Thanks for the comment. We chose to maintain the order of the subchapters because we understand that we must first bring the fruit data.
L 185 Footnote table 1, the values must be expressed g/100 gr FW (fresh weight)
Thanks for the comment. Corrected in the text.
Figure 1. L 223 To correct the P values for the confidence interval
Thanks for the comment. Corrected in the text.
Figure 1. A- How do you explain values over 100% for control in the case of cell viability under treatment with yellow acacia?
Thanks for the comment. The results, even the control, are expressed as a mean with standard deviation variation. It is normal for some data to vary more above and below the average. Maybe that's why the data appears to be below 100% or above 100%. We review all results to confirm.
Fig 2.C – How do you explain the control value of less than 100% ??
Thanks for the comment. The results, even the control, are expressed as a mean with standard deviation variation. It is normal for some data to vary more above and below the average. Maybe that's why the data appears to be below 100% or above 100%. We review all results to confirm.
L 247-250 Correct and add information regarding 2B, D and F.
Thanks for the comment. Corrected in the text.
L 250. Correct the value p<0.05
Thanks for the comment. Corrected in the text.
L 276. n represents the number of repetitions used to determine MIC/ MFC for each of the Candida isolates. Correct as required.
Thanks for the comment. Corrected in the text.
Table 2. The maximum concentration of fluconazole taken in the study is 32mg/ml, according to materials and methods. To be corrected in Table 2 and footnote.
Thanks for the comment. Corrected in the text.

Round 2
Reviewer 1 Report
Comments and Suggestions for Authors
Overall, the suggestions made were considered.
Still, "ºC" should be corrected thoroughly the manuscripts (e.g. Lines 94, 95, 105...)